# Collision rate ansatz for quantum integrable systems

Takato Yoshimura[*,♠] and Herbert Spohn[⋆,*]

[*] *Department of Physics, Tokyo Institute of Technology, Ookayama 2-12-1, Tokyo 152-8551, Japan*
[♠] *Institut de Physique Théorique Philippe Meyer, École Normale Supérieure, PSL University, Sorbonne Universités, CNRS, 75005 Paris, France*
[⋆] *Physik Department and Zentrum Mathematik, Technische Universität München, Boltzmannstrasse 3, 85748 Garching, Germany*

For quantum integrable systems the currents averaged with respect to a generalized Gibbs ensemble are revisited. An exact formula is known, which we call "collision rate ansatz". While there is considerable work to confirm this ansatz in various models, our approach uses the symmetry of the current-charge susceptibility matrix, which holds in great generality. Besides some technical assumptions, the main input is the availability of a self-conserved current, i.e. some current which is itself conserved. The collision rate ansatz is then derived. The argument is carried out in detail for the Lieb-Liniger model and the Heisenberg XXZ chain. The Fermi-Hubbard is not covered, since no self-conserved current seems to exist. It is also explained how from the existence of a boost operator a self-conserved current can be deduced.

July 29, 2020

# 1 Introduction

Hydrodynamics is a universal tool to describe the long-wavelength dynamics of many-body systems, both quantum and classical. The cornerstone of hydrodynamics is the assumption of local equilibrium and its stable propagation in spacetime. Thereby the complex dynamics of a many-body system is guided by interactions between conserved charges only [1]. As a consequence, the dynamics is determined by a coupled set of continuity equations for the average charge densities and currents. Such a system closes only if all local conservation laws are included. For a generic system one expects to have a few of them, hence the description only involves a few coupled hyperbolic conservation laws. But for integrable dynamics the conserved fields are labelled by a spectral parameter from the real line, or even larger sets, depending on the model. Such generalized hydrodynamics (GHD) is particularly useful for quantum integrable systems, for which, even numerically, tracing the late-time dynamics is notoriously difficult mainly due to the rapid increase of entanglement across distant spatial regions [2–4].

The hydrodynamics of integrable systems was accomplished in 2016 [5,6], giving rise to a flux of related studies [7–20], including the determination of Drude weights [21], Green-Kubo type formulas for the transport coefficients [22,23], and applications to classical integrable systems [24–27]. On the ballistic spacetime scale GHD turns out to have a particularly simple structure, since charge densities and currents evaluated with respect to a generalized Gibbs ensemble (GGE) [28] can be written in terms of the thermodynamic Bethe ansatz (TBA), which is already known as a systematic method in the study of thermodynamics of quantum integrable systems. Compared to statics the novel key element is the *effective velocity* $v^{\mathrm{eff}}(\theta)$ as a function of the rapidity $\theta$[1]. This quantity describes the velocity of quasiparticles at the hydrodynamic scale and thereby expresses current densities as a nonlinear functional of the charge densities. The functional form of $v^{\mathrm{eff}}$ was first conjectured in [5,6]. In the former one finds a sketchy reasoning as well as an argument from the crossing symmetry in case of relativistic field theories with diagonal scattering. The effective velocity is written as the solution of a rate equation counting the number of collisions per unit time experienced by a single tracer quasiparticle in a fluid of quasiparticles distributed according to some GGE. In this article we use the notion *collision rate ansatz*, as reflecting the physics intuition behind the defining formula for $v^{\mathrm{eff}}$.

While the formula for $v^{\mathrm{eff}}$ was rapidly adopted, satisfactory theoretical arguments have become available only recently: the form factor expansion is used to establish the collision rate ansatz for both diagonally-scattering relativistic field theories [30] and the XXZ spin-$\frac{1}{2}$ chain [31]. It was also demonstrated that thermodynamic form factor expansion[2], a slight generalization of the standard form factor expansion, yields the collision rate ansatz in

---

[1]The same quantity appeared in a different context even before [29], where no specific name of the quantity was given.

[2]The rigour of this approach, however, is still largely missing and under development [32].

the Lieb-Liniger model [23]. A distinct approach is used in [33], where the collision rate ansatz is confirmed for the models solvable by nested Bethe ansatz by employing a relation derived from long-range deformations of the chain. A generalization of the collision rate ansatz to excited states has been accomplished in [34]. In addition, proofs for some specific cases are available as well. For instance, in [35] the exact form of the current operators and their averages are derived rigorously for the free fermion chain. The collision rate ansatz for the spin current of the XXZ spin-$\frac{1}{2}$ chain is established in [36]. As obtained very recently [37], the validity of the collision rate ansatz can be directly checked from the exact current operators in the XXZ spin-$\frac{1}{2}$.

The aim of this manuscript is to add a very different line of arguments for justifying the collision rate ansatz in interacting quantum integrable systems. In fact, our argument is more direct and resorts neither to form factor expansions nor deformations. Our method is based on the availability of a self-conserved current. By this we mean a current, which itself appears in the list of conserved charges. For the Lieb-Liniger model the particle current is momentum which is itself conserved. Also, as well-known, in the XXZ spin-$\frac{1}{2}$ chain the energy current is self-conserved. However, for the Fermi-Hubbard model the energy current is not conserved [38] and possibly the model has no self-conserved current at all. More generally, the availability of a self-conserved current is ensured by the existence of an algebra involving conserved charges and the boost operator, which can be written as the first moment of some conserved charge density [39–41]. Such algebra is directly linked to the global symmetry of the model (e.g. Lorentz or Galilean invariance) in continuum systems, while in lattice systems it can be thought of as the lattice generalization of the Lorentz algebra. Thus our result could be rephrased that the existence of the algebra-generating boost operator alone suffices to validate the collision rate ansatz. It turns out that our approach can be naturally extended to prove the collision rate ansatz for currents associated to flows generated by higher conserved charges. Details will be presented in Appendix A.

## 2  Collision rate ansatz for integrable field theories

Integrable quantum systems have an extensive number of (quasi-)local conserved charges. To simplify, we shall focus on the case of single quasi-particle species with diagonal scatterings. The more complicated structure of the XXZ model will be discussed in Sect. 3. The charges are denoted by $Q_j = \int \mathrm{d}x\, \mathfrak{q}_j(x)$, $j = 0, 1, \dots$, in particular $[H, Q_j] = 0$. In the generalized Gibbs ensemble (GGE) each one of them is controlled by a chemical potential, $\mu_j$, and the corresponding density matrix reads

$$\rho_{\mathrm{GGE}} = \frac{e^{-\sum_j \mu_j Q_j}}{\mathrm{Tr}(e^{-\sum_j \mu_j Q_j})}. \tag{1}$$

From the field theory under consideration one has given a dispersion relation $E(\theta)$ as a function of the rapidity $\theta$ with the momentum $p(\theta) = E'(\theta)$. In fact, our the argument

will be written out in detail for the Lieb-Liniger, a Galilei-invariant field theory, but with a notation which will make the application to other field theories straightforward. We recall that for the Lieb-Liniger model $E(\theta) = \frac{1}{2}\theta^2$ in units for which the bare particle mass $m = 1$. Furthermore given is the two-body scattering matrix $S(\theta, \vartheta)$, in terms of which the two-particle differential scattering kernel is given by

$$T(\theta, \vartheta) = -\mathrm{i}\frac{1}{2\pi}\partial_\theta \log S(\theta, \vartheta). \tag{2}$$

The free energy of the system can be computed from the TBA equations

$$\varepsilon(\theta) = \sum_{j=0}^{\infty} \mu_j h_j(\theta) - \int_{\mathbb{R}} \mathrm{d}\vartheta\, T(\theta, \vartheta) \log(1 + e^{-\varepsilon(\vartheta)}) \tag{3}$$

with $h_j(\theta)$ the one-particle eigenvalue associated to the charge $Q_j$, $h_j(\theta) = \theta^j$ in our case. From the pseudo-energy $\varepsilon$ one obtains the occupation function $n(\theta) = 1/(1 + e^{\varepsilon(\theta)}) = \rho(\theta)/\rho^{\mathrm{tot}}(\theta)$ with $\rho$ the density of particles and $\rho^{\mathrm{tot}}$ the density of states, related through

$$\rho^{\mathrm{tot}}(\theta) = \frac{1}{2\pi}p'(\theta) + \int_{\mathbb{R}} \mathrm{d}\vartheta\, T(\theta, \vartheta)\rho(\vartheta). \tag{4}$$

In terms of these quantities, the GGE average of a charge density, $\mathsf{q}[h_j] := \langle \mathsf{q}_j(0)\rangle_{\mathrm{GGE}}$, can be written as [43]

$$\mathsf{q}[h_j] = \langle \rho h_j \rangle = \frac{1}{2\pi}\langle p' n h_j^{\mathrm{dr}}\rangle, \tag{5}$$

Here, for any function $f(\theta)$ we use the shorthand $\langle f \rangle = \int_{\mathbb{R}} \mathrm{d}\theta\, f(\theta)$. The dressing transformation is defined through

$$f^{\mathrm{dr}}(\theta) = \big((1 - Tn)^{-1}f\big)(\theta). \tag{6}$$

In the context of GHD it was a major discovery that the current average also admits a similar TBA expression [5,6]. The microscopic current is defined through the continuity equation $\partial_t \mathsf{q}_j(x,t) + \partial_x \mathsf{j}_j(x,t) = 0$. Then the time $t = 0$ total current is given by $J_j = \int \mathrm{d}x\, \mathsf{j}_j(x,0)$ and the corresponding GGE average equals $\mathsf{j}[h_j] = \langle \mathsf{j}_j(0,0)\rangle_{\mathrm{GGE}}$. Since $\mathsf{j}[h_j]$ is linear in $h_j$, in analogy to (5) one starts from the ansatz

$$\mathsf{j}[h_j] = \langle \rho v^{\mathrm{eff}} h_j \rangle = \frac{1}{2\pi}\langle E' n h_j^{\mathrm{dr}}\rangle \tag{7}$$

with the effective velocity $v^{\mathrm{eff}}$ given as solution of the rate equation

$$v^{\mathrm{eff}}(\theta) = \frac{E'(\theta)}{p'(\theta)} - 2\pi \int_{\mathbb{R}} \mathrm{d}\vartheta \frac{T(\theta, \vartheta)}{p'(\theta)}\rho(\vartheta)(v^{\mathrm{eff}}(\theta) - v^{\mathrm{eff}}(\vartheta)). \tag{8}$$

Its physical interpretation has been mentioned already, but can now be stated more precisely. $\theta$ is the spectral parameter of the tracer quasi-particle, which is moving in a fluid characterized by the density $\rho(\vartheta)$. The bare velocity of the tracer particle is $E'/p'$, which

is modified through collisions with fluid particles. Under integral the first factor is the jump size of either sign and the second factor is the number of collisions per unit time [10]. Our task is to establish the ansatz (8) on the basis of a given microscopic model, for which purpose a convenient form of the effective velocity is

$$v^{\text{eff}}(\theta) = \frac{(E')^{\text{dr}}(\theta)}{(p')^{\text{dr}}(\theta)}. \tag{9}$$

The first input to our proof are the charge-charge and current-charge susceptibility matrices which are defined by [21]

$$C_{ij} = \int_{\mathbb{R}} dx \langle \mathsf{q}_i(x,0)\mathsf{q}_j(0,0)\rangle^{\text{c}}_{\text{GGE}} = -\frac{\partial}{\partial \mu_j}\mathsf{q}_i, \tag{10}$$

$$B_{ij} = \int_{\mathbb{R}} dx \langle \mathsf{j}_i(x,0)\mathsf{q}_j(0,0)\rangle^{\text{c}}_{\text{GGE}} = -\frac{\partial}{\partial \mu_j}\mathsf{j}_i \tag{11}$$

with the superscript referring to connected correlation functions. The matrix $C$ is symmetric by construction. Less obvious, but also $B$ is symmetric. Making use of the conservation laws, spacetime stationarity, and clustering of connected correlation functions [5, 23], one arrives at

$$\langle \mathsf{j}_i(x,t)\mathsf{q}_j(0,0)\rangle^{c}_{\text{GGE}} = \langle \mathsf{j}_j(x,t)\mathsf{q}_i(0,0)\rangle^{c}_{\text{GGE}}, \tag{12}$$

which implies the symmetry of $B$. In fact the symmetry holds in more general situations, and the precise condition of its validity is discussed in [44].

The second input is the existence of a self-conserved current. For the Lieb-Liniger model $J_0 = Q_1$, hence $J_0$, which is the total current associated to the particle number operator $Q_0 = N$, is self-conserved. As will be discussed, for other models there might be a different self-conserved current. The symmetry of $B$ yields then the following nontrivial identity

$$\partial_{\mu_1}\mathsf{q}_j = \partial_{\mu_0}\mathsf{j}_j. \tag{13}$$

Next note that by linearity in $h_j$ the left identity of (7) still holds provided $v^{\text{eff}}(\theta)$ is replaced by the yet unknown current density $\bar{v}(\theta)$. Therefore (13) becomes

$$\int_{\mathbb{R}} d\theta h_j(\theta)\partial_{\mu_0}\big(\rho(\theta)\bar{v}(\theta)\big) = \int_{\mathbb{R}} d\theta h_j(\theta)\partial_{\mu_1}\rho(\theta), \tag{14}$$

satisfied for all $j$. Since the space spanned by $h_j$'s is complete one arrives at the pointwise identity

$$\partial_{\mu_0}\big(\rho\bar{v}\big) = \partial_{\mu_1}\rho. \tag{15}$$

To be shown is $\bar{v} = v^{\text{eff}}$.

From differentiating the TBA equations with respect to $\mu_0, \mu_1$ the relations

$$\partial_{\mu_0}n = -n(1-n)(p')^{\text{dr}}, \quad \partial_{\mu_1}n = -n(1-n)(E')^{\text{dr}}, \tag{16}$$

hold and imply

$$\partial_{\mu_1}(p')^{\mathrm{dr}} = \partial_{\mu_0}(E')^{\mathrm{dr}}. \tag{17}$$

Using

$$\rho v^{\mathrm{eff}} = \frac{1}{2\pi}n(E')^{\mathrm{dr}}, \tag{18}$$

one arrives at

$$\partial_{\mu_0}(\rho v^{\mathrm{eff}}) = \frac{1}{2\pi}\partial_{\mu_0}((E')^{\mathrm{dr}}n) = \frac{1}{2\pi}\left((E')^{\mathrm{dr}}\partial_{\mu_0}n + n\partial_{\mu_0}(E')^{\mathrm{dr}}\right)$$

$$= \frac{1}{2\pi}\left((p')^{\mathrm{dr}}\partial_{\mu_1}n + n\partial_{\mu_1}(p')^{\mathrm{dr}}\right) = \frac{1}{2\pi}\partial_{\mu_1}((p')^{\mathrm{dr}}n) = \partial_{\mu_1}(\rho^{\mathrm{tot}}n) = \partial_{\mu_1}\rho. \tag{19}$$

Altogether we obtained $\partial_{\mu_0}\left(\rho(\bar{v} - v^{\mathrm{eff}})\right) = 0$. Hence $\rho(\bar{v} - v^{\mathrm{eff}})$ is pointwise constant in $\mu_0$. From the TBA equation (3) one infers that the pseudo-energy $\varepsilon(\theta) \simeq \mu_0$ for $\mu_0 \to \infty$, thus $n(\theta) = 1/(1 + e^{\varepsilon(\theta)}) \to 0$. Since $\rho^{\mathrm{tot}}(\theta)$ is uniformly bounded in $\mu_0$, also $\rho$ vanishes. Physically one would expect that $\bar{v}$ is locally bounded for large $\mu_0$ and hence $\rho\bar{v} \to 0$ pointwise when $\mu_0 \to \infty$. We need this property as an additional assumption. If so, we conclude that the free constant must be zero, establishing

$$\bar{v} = v^{\mathrm{eff}}. \tag{20}$$

The only property needed for the above argument is the existence of a self-conserved current. In Galilei-invariant theories with the particle number conservation, the number current equals the momentum and our requirement is satisfied. Thereby our argument can be extended to other Galilei-invariant theories such as the Gaudin-Yang model [47], which is solved by a nested Bethe ansatz. In relativistic field theories Lorentz invariance ensures a distinct self-conserved current. In this case, the energy current $J_2$ coincides with the momentum operator $Q_1$, from which $\partial_{\mu_1}\mathsf{q}_j = \partial_{\mu_2}\mathsf{j}_j$ follows. In single-species models with diagonal-scatterings, the dispersion relation has the relativistic form $p(\theta) = m\sinh\theta, E(\theta) = m\cosh\theta$, $m$ the particle mass, and the task then boils down to show

$$\partial_{\mu_2}(\rho v^{\mathrm{eff}}) = \partial_{\mu_1}\rho. \tag{21}$$

This can be confirmed in a similar fashion as above by noting that $h_2^{\mathrm{dr}}(\theta) = E^{\mathrm{dr}}(\theta) = 2\pi\rho^{\mathrm{tot}}(\theta)$. We therefore conclude

$$\partial_{\mu_2}\left(\rho(\bar{v} - v^{\mathrm{eff}})\right) = 0. \tag{22}$$

Since $h_2(\theta) = m\cosh\theta > 0$, this time in the $\mu_2 \to \infty$ limit $n(\theta) \to 0$. Assuming a similar behavior of $\bar{v}$ as in the Lieb-Liniger model, i.e. $\rho\bar{v} \to 0$ when $\mu_2 \to \infty$, we finally have $\bar{v} = v^{\mathrm{eff}}$.

As in the non-relativistic cases, the above argument for relativistic theories can be straightforwardly generalized to other relativistic models such as the sine-Gordon model and the $O(N)$ non-linear sigma model, in which cases the energy current is quasi-conserved. However, the situation is different for integrable spin chains, which do not possess an evident continuous symmetry implying the existence of a self-conserved current. As discussed in Sect. 4, the boost operator could be useful tool in finding such a current.

# 3 Collision rate ansatz for the XXZ spin-$\frac{1}{2}$ chain

For the XXZ spin-$\frac{1}{2}$ chain the energy current is self-conserved. Because of strings in the Bethe equations the structure of the charges is more involved than for Lieb-Liniger. Thus the model is an interesting test for our method.

The hamiltonian of the XXZ model reads

$$H = J \sum_{n \in \mathbb{Z}} (S_n^x S_{n+1}^x + S_n^y S_{n+1}^y + \Delta S_n^z S_{n+1}^z), \tag{23}$$

where we set $J = 1$. The TBA structure of the chain strongly depends on the value of $\Delta$ [48] and, as a consequence, the Drude weight changes sensitively with the isotropy parameter $\Delta$ [6, 45, 46]. However the energy current is self-conserved for any value of $\Delta$, which is the only requirement for our argument to work. For concreteness, we focus on the gapless regime here ($|\Delta| < 1$), but the gapped regime can be handled in a similar fashion.

The structure of TBA for the gapless XXZ spin-$\frac{1}{2}$ chain can be arranged so as to become rather similar to that for the Lieb-Liniger model. This is achieved by choosing particular values of $\Delta$, which are called roots of unity,

$$\Delta = \cos \omega, \quad \frac{\omega}{\pi} = \frac{1|}{|\nu_1} + \frac{1|}{|\nu_2} + \cdots + \frac{1|}{|\nu_{\bar{\ell}}} \tag{24}$$

with $\bar{\ell}$ the length of the continued fraction and some positive integers $\nu_1, \cdots, \nu_{\ell-1} \geq 1$, $\nu_{\ell} \geq 2$, $\ell = 1, ..., \bar{\ell}$, using Pringsheim's notation. The number of strings equals $\mathsf{s} = \sum_{\ell=1}^{\bar{\ell}} \nu_\ell$, hence finite for such a $\Delta$. The set of string labels is denoted by $\mathbb{S} = \{1, ...., \mathsf{s}\}$. The resulting TBA equations now involve various types of strings [48]. Apart from the fact that there are more particle types (strings), as a further modification of the TBA equations, the overall sign of $p'_j(\lambda)$ depends on the type $j \in \mathbb{S}$. This is a consequence of a reparametrization of rapidities so as to make the differential scattering kernel $T_{jk}(\lambda)$ symmetric, which in turn induces a change to the integration measure $\int \mathrm{d}\lambda \mapsto \sum_j \sigma_j \int \mathrm{d}\lambda$ [23]. Here $\sigma_j = \mathrm{sign}(q_j)$, where $q_j$ is related to the parity of $j$-th string and depends on $\Delta$ [48].

In the gapless phase of XXZ, $\rho_j^{\mathrm{tot}}(\lambda)$ is given by $\rho_j^{\mathrm{tot}}(\lambda) = \sigma_j (p'_j)^{\mathrm{dr}}(\lambda)/(2\pi)$, where for any function $f_j(\lambda)$ the dressing transformation is defined by

$$f_j^{\mathrm{dr}}(\lambda) = f_j(\lambda) - \sum_{k \in \mathbb{S}} \sigma_k \int_{\mathbb{R}} \mathrm{d}\vartheta T_{jk}(\lambda - \vartheta) n_k(\vartheta) f_k^{\mathrm{dr}}(\vartheta). \tag{25}$$

Note that by convention the sign of $T$ is opposite to the one used in the previous section. It will be convenient to work with integral operators. They act on functions over $\mathbb{R} \times \mathbb{S}$, which are equipped with the standard scalar product

$$\langle f, g \rangle = \sum_{j \in \mathbb{S}} \int_{\mathbb{R}} \mathrm{d}\lambda f_j(\lambda) g_j(\lambda). \tag{26}$$

Then, employing integral operators, (25) becomes

$$f^{\mathrm{dr}} = (1 + Tn\sigma)^{-1}f = \sigma(\sigma + Tn)^{-1}f. \tag{27}$$

To be complete, in the gapless phase of the XXZ spin-$\frac{1}{2}$ chain the bare momentum $p_j(\lambda)$ and the differential scattering kernel $T_{jk}(\lambda - \mu)$ are

$$p_j(\lambda) = p_{n_j}(\lambda|v_j) = 2v_j \tan^{-1}\left[(\cot\frac{n_j\omega}{2})^{v_j}\tanh\frac{\lambda}{2}\right], \tag{28}$$

$$T_{jk}(\lambda) = \frac{1}{2\pi}\big[p'_{|n_j-n_k|}(\lambda|v_jv_k) + 2p'_{|n_j-n_k|+2}(\lambda|v_jv_k) \\ + \cdots + 2p'_{|n_j+n_k|-2}(\lambda|v_jv_k) + p'_{|n_j+n_k|}(\lambda|v_jv_k)\big]. \tag{29}$$

where $n_j$ and $v_j$ are the length and the parity of the $j$-th string. For instance, when $\omega = \pi/\nu$ with $2 \le \nu \in \mathbb{N}$, those $n_j, q_j, v_j$'s are given by [48]

$$\begin{cases} n_j = j, & v_j = 1, & q_j = \nu - n_j, & j = 1, 2, \ldots, \nu - 1, \\ n_\nu = 1, & v_\nu = -1, & q_\nu = -1, & j = \nu. \end{cases} \tag{30}$$

At the roots of unity, there is a family of quasi-local conserved charges $Q_n^{(s)}$ labeled by integers $n \in \mathbb{N}$ and half-integer $s \in \frac{1}{2}\mathbb{N}$, which corresponds to the higher-spin representation of $\mathcal{U}_q(\mathfrak{sl}(2))$ [50]. The energy current is $Q_2^{(1/2)}$, hence conserved. Writing the one-particle eigenvalue of the charges as $h_{n,j}^{(s)}(\lambda)$, the GGE average of charge and currents densities take a form similar to the continuum case,

$$\mathsf{q}[h] = \langle h, \rho \rangle = \frac{1}{2\pi}\langle h, n(\sigma + Tn)^{-1}p' \rangle, \tag{31}$$

$$\mathsf{j}[h] = \langle h, v^{\mathrm{eff}}\rho \rangle = \frac{1}{2\pi}\langle h, n(\sigma + Tn)^{-1}E' \rangle. \tag{32}$$

Now, let us consider the energy current $\mathsf{j}[E]$, where $E = h_1^{(1/2)}$. Using $h_2^{(1/2)} = -\frac{1}{2}(\sin\omega)E'$ and $E = -\frac{1}{2}(\sin\omega)p'$ [6], it follows that

$$\mathsf{j}[E] = \frac{1}{2\pi}\langle E, n(\sigma + Tn)^{-1}E' \rangle = \langle p', n(\sigma + Tn)^{-1}h_2^{(1/2)} \rangle \\ = \langle h_2^{(1/2)}, n(\sigma + Tn)^{-1}p' \rangle = \mathsf{q}[h_2^{(1/2)}], \tag{33}$$

which is in agreement with $Q_2^{(1/2)} = J_E$ and also implies $\mathsf{q}[E'] = \mathsf{j}[p']$. Having these relations at our disposal, let us proceed to the proof. As in the field theory case, (13) and its consequence (15) are the key identities. The only difference to these identities is that the self-conserved current is $J_1^{(1/2)} = Q_2^{(1/2)}$. Denoting the Lagrange multipliers associated to $Q_2^{(1/2)}$ and $Q_1^{(1/2)}$ by $\mu_2$ and $\mu_1$, respectively, we notice first that

$$\rho^{\mathrm{tot}}\partial_{\mu_2}n = -n(1-n)(h_2^{(1/2)})^{\mathrm{dr}}\rho^{\mathrm{tot}} = \tfrac{1}{2}(\sin\omega)n(1-n)(E')^{\mathrm{dr}}\frac{(p')^{\mathrm{dr}}}{2\pi\sigma} \\ = -\tfrac{1}{2\pi}n(1-n)\sigma(E')^{\mathrm{dr}}E^{\mathrm{dr}} = \tfrac{1}{2\pi}\sigma(E')^{\mathrm{dr}}\partial_{\mu_1}n, \tag{34}$$

which implies $\partial_{\mu_2}(p')^{\mathrm{dr}} = \partial_{\mu_1}(E')^{\mathrm{dr}}$. As a final step,

$$\partial_{\mu_2}\rho = \rho^{\mathrm{tot}}\partial_{\mu_2}n + n\partial_{\mu_2}\rho^{\mathrm{tot}} = \frac{\sigma}{2\pi}\left((E')^{\mathrm{dr}}\partial_{\mu_1}n + n\partial_{\mu_1}(E')^{\mathrm{dr}}\right) = \partial_{\mu_1}(\rho v^{\mathrm{eff}}), \qquad (35)$$

which then yields $\partial_{\mu_1}[\rho_j(\lambda)(\bar{v}_j(\lambda) - v_j^{\mathrm{eff}}(\lambda))] = 0$.

For given $j$ the energy one-particle eigenvalue equals $E_j(\lambda) = -\frac{1}{2}(\sin\omega)p'_j(\lambda)$ with the property that either $E_j(\lambda) > 0$ or $E_j(\lambda) < 0$. In the former case, we let $\mu_1 \to \infty$. From the TBA it follows that $n_j(\lambda) \to 0$ in this limit. In the latter case we let $\mu_1 \to -\infty$ and, as before, conclude that $n_j(\lambda) \to 0$. Finally, once again assuming a similar behavior of $\bar{v}_j$ as in the previous continuum cases, i.e. $\rho_j\bar{v}_j \to 0$ for each string $j$ under either $\mu_1 \to -\infty$ or $\mu_1 \to \infty$, the free constant vanishes and $\rho_j(\lambda)(\bar{v}_j(\lambda) - v_j^{\mathrm{eff}}(\lambda)) = 0$ for all $j, \lambda$, implying the desired result, $\bar{v}_j(\lambda) = v_j^{\mathrm{eff}}(\lambda)$.

# 4  Boost operator in spin chains

As we saw in the previous section, the existence of a self-conserved current gives rise to the collision rate ansatz. One then might wonder how such a current can be obtained in general. Indeed, even without invoking integrability, the existence of a self-conserved current can be directly inferred from either Galilei or Lorentz symmetry of the quantum field theory under consideration. This is no longer true for spin chains where such a continuous symmetry is absent and a self-conserved current has to be found along an alternative route. An essential tool for this task turns out to be the boost operator. First we briefly recall its basic property in the context of XYZ spin-$\frac{1}{2}$ chain $H = \sum_{j\in\mathbb{Z}}h(j)$, where

$$h(j) = -\frac{1}{2}(J_x S_j^x S_{j+1}^x + J_y S_j^y S_{j+1}^y + J_z S_j^z S_{j+1}^z). \qquad (36)$$

A tower of conserved charges can be systematically obtained by the row-to-row transfer matrix as

$$\log T(\lambda) = \sum_{n=0}^{\infty}\frac{\lambda^n}{n!}Q_n, \qquad (37)$$

hence

$$Q_n = \frac{\mathrm{d}^n}{\mathrm{d}\lambda^n}\log T(\lambda)\Big|_{\lambda=0}. \qquad (38)$$

Let us consider some operator $\mathcal{O}$ which is constructed from a local density $o(j)$ through $\mathcal{O} = \sum_{j\in\mathbb{Z}}o(j)$. Then the boost operator is defined through

$$K[\mathcal{O}] = \sum_{j\in\mathbb{Z}}jo(j). \qquad (39)$$

The boost operator associated to the Hamiltonian[3] $K[H] = \sum_{j \in \mathbb{Z}} j h(j)$ indeed generates a boost, which is evident from the commutation relation with the transfer matrix

$$[K[H], T(\lambda)] = \partial_\lambda T(\lambda), \tag{40}$$

which in turn amounts to [39, 40]

$$[K[H], Q_n] = \mathrm{i} Q_{n+1}. \tag{41}$$

The fact that the boost operator $K[H]$ generates the conserved charges recursively bears momentous implications. We recall the continuity equation in spin chains

$$\mathrm{i}[H, \mathfrak{q}_n(j)] = \mathfrak{j}_n(j) - \mathfrak{j}_n(j+1). \tag{42}$$

Multiplying $j$ to both sides and summing over $j$, we formally obtain

$$\mathrm{i}[H, K[Q_n]] = \sum_{j \in \mathbb{Z}} \mathfrak{j}_n(j). \tag{43}$$

As was remarked in [33] the relation (43) is only formal, and is in general plagued by the divergence stemming from the charge density with an infinitely large coefficient. Nevertheless such a divergence can always be circumvented by subtracting a conserved charge $Q_n$ with a correspondingly diverging prefactor.

In spin chains, it is conventional to choose $Q_0 = N = \sum_n S_n^z$ and $Q_1 = H$. Then, choosing $n = 1$ in (41) and (43), we observe that $J_1 = \sum_{j \in \mathbb{Z}} \mathfrak{j}_1(j)$ is a self-conserved current, i.e.

$$Q_2 = \sum_{j \in \mathbb{Z}} \mathfrak{j}_1(j). \tag{44}$$

Note that the above construction of a self-conserved current suggests $J_1$ being actually the only self-conserved current under the Hamiltonian flow.

In fact, the recursive commutation relations (41) and (43) can be thought of as the lattice analogue of the Poincaré algebra. Indeed, in the continuum limit the XYZ spin chain becomes the relativistic massive Thirring/sine-Gordon model [40]. The upshot of this limit is that the first few commutation relations (41) reduce to the usual Poincaré algebra in (1+1)-dimension, which is closed in itself,

$$[H, P] = 0, \quad [K[H], H] = \mathrm{i} P, \quad [K[H], P] = \mathrm{i} H. \tag{45}$$

Using (43) this implies $J_E = P$, which is what one would expect from Lorentz invariance.

Naturally one can further take the non-relativistic limit of the Poincaré algebra, which is nothing but the Galilean algebra. In particular, when the resulting theory has $U(1)$-symmetry (e.g. conserves particle number), such as the Lieb-Liniger model, the Galilean algebra is centrally extended to the Bargmann algebra whose commutation relations read

$$[H, P] = 0, \quad [K[N], H] = \mathrm{i} P, \quad [K[N], P] = \mathrm{i} N, \tag{46}$$

---

[3]We shall call it simply "the boost operator" unless otherwise stated.

where $N = Q_0$ is the $U(1)$ charge. This algebra then entails $J_0 = P$, which again is merely a consequence of Galilean invariance.

So far we have demonstrated that the XYZ spin-$\frac{1}{2}$ chain, hence also the XXZ spin-$\frac{1}{2}$ chain, possesses a self-conserved current $\mathsf{j}_1$ thanks to the boost operator that satisfies the properties specified above. This is also true for other integrable spin chains, provided that there is a boost operator which satisfies (41) and (43). A natural question is then, whether there are integrable systems which for some reason fail to have a boost operator of the form (39)? The answer is yes, and a notable example is the Fermi-Hubbard model (FHM), for which the energy current is not conserved [38]. This is consistent with the fact that FHM does not have the standard boost operator, and the lack of it suggests that there could be no self-conserved current at all. In fact, at the root of the existence of such a boost operator is the lattice Lorentz invariance of the system whose algebra is given by the ladder commutation relations (41) [40]. The invariance under a lattice Lorentz boost manifests itself through the $R$-matrix of the system being of the form $R(\lambda, \mu) = R(\lambda - \mu)$, hence invariant under a boost. The lattice Lorentz invariance reduces to the standard continuum Lorentz invariance in the continuum limit. FHM does not allow such invariance, since the model does not admit any continuum limit under which Lorentz invariance is achieved. Indeed, the low-energy physics of FHM is not a Luttinger liquid, but instead charges and spin carry gapless excitations with different velocities, which implies that the physics depends on the frame.

This being said, it is actually possible to define a slightly generalized boost operator in FHM, which still satisfies (41) [49]. As a caveat, the generalized boost operator is not exactly the same as (39) and the connection to the conservation laws (42) is lost. Therefore a self-conserved current in FHM, if it should exist, has to be looked for by other means.

# 5    Conclusions

In this article, we proved the collision rate ansatz for a wide class of quantum integrable systems, once the existence of a self-conserved current is ensured. It turns out that the existence of such a self-conserved current is directly linked to the boost operator, which is written as the first moment of some charge density. When such a boost operator forms an algebra with conserved charges, the continuity equation and the algebra immediately give a self-conserved current. In fact, the construction of a self-conserved current can be immediately extended to the generalized currents describing the flow of other conserved charges. Generalized currents $\mathsf{j}_{nm}(j)$ associated to $Q_n$ are defined by the continuity equation [16, 33]

$$\mathrm{i}[Q_m, \mathfrak{q}_n(j)] = \mathsf{j}_{nm}(j) - \mathsf{j}_{nm}(j+1) \tag{47}$$

for each flow generated by $Q_m$. Of course, $m = 1$ corresponds to the standard Hamiltonian flow. We then find that along each $m$-th flow there is always a self-conserved total current

$\sum_{j \in \mathbb{Z}} \mathbf{j}_{1m}(j) = J_{1m} = Q_{m+1}$: the collision rate ansatz for such generalized currents are provided in the appendix A. Indeed, such a boost operator has been used to implement long-range deformations of integrable spin chains, from which the finite-volume diagonal matrix elements of current operators were obtained [33]. It would be very interesting to figure out the connection between the use of the boost operator in our proof and the one in [33], with the hope to better understand the overarching role of the boost operator in GHD.

Finally, let us remark that the approach followed here is in spirit the same as the one for the classical Toda lattice [51]. In this model the stretch current equals the negative of the momentum, hence is indeed conserved. We expect that our approach will be applicable to a larger variety of integrable systems, both quantum and classical, provided that the system has a self-conserved current.

# 6 Ackowledgement

TY is grateful to Enej Ilievski for useful comments on the Fermi-Hubbard model, and in particular informing him a paper [49], in which the (generalized) boost operator in the model is discussed. HS thanks Tomohiro Sasamoto for his generous hospitality at Tokyo Institute of Technology.

# A Generalized current

In this appendix, we shall demonstrate that the proof we presented in the main text can be readily generalized to generalized currents, which are defined by the generalized Heisenberg equation (47). Below, focusing on the case of the XXZ spin-$\frac{1}{2}$ chain, we derive the collision rate formula for the generalized current $\mathbf{j}_{jk}$:

$$\mathbf{j}_{jk} := \langle \mathbf{j}_{jk}(0) \rangle_{\text{GGE}} = \sum_{a \in \mathbb{S}} \int_{\mathbb{R}} \frac{\mathrm{d}\theta}{2\pi} \sigma_a (h'_{k,a})^{\mathrm{dr}} n_a h_{j,a}. \tag{48}$$

In what follows let us suppress the string indices and the summation over them, which play no role in the proof. First we note that $B_{ijk} := \frac{\partial}{\partial \beta^i} \mathbf{j}_{jk}$ satisfies $B_{ijk} = B_{jik}$, which can be shown as in the usual current-charge susceptibility matrix $B_{ij}$ [52]. As remarked in the conclusion, the generalized bridging pair $J_{1m} = Q_{m+1}$ exists in the XXZ chain. This implies that we have an identity $B_{j1k} = B_{1jk} = C_{j,k+1}$, i.e.

$$\frac{\partial}{\partial \beta^1} \mathbf{j}_{jk} = \frac{\partial}{\partial \beta_{k+1}} \mathbf{q}_j. \tag{49}$$

From the continuity equation, we can again assume that $\mathbf{j}_{jk}$ should be of the following form

$$\mathbf{j}_{jk} = \int \frac{\mathrm{d}\theta}{2\pi} \sigma \bar{h}_k n h_j. \tag{50}$$

The remaining task is then to show $\bar{h}_k = (h'_k)^{\mathrm{dr}}$. For that we establish

$$\frac{1}{2\pi}\frac{\partial}{\partial\beta^1}(\sigma(h'_k)^{\mathrm{dr}}n) = \frac{\partial}{\partial\beta_{k+1}}\rho, \tag{51}$$

which is a simple matter to check. First,

$$\rho^{\mathrm{tot}}\frac{\partial}{\partial\beta_{k+1}}n = -\sigma\frac{(p')^{\mathrm{dr}}}{2\pi}(1-n)h^{\mathrm{dr}}_{k+1} = -\alpha\sigma\frac{(p')^{\mathrm{dr}}}{2\pi}(1-n)(h'_k)^{\mathrm{dr}}$$

$$= -\sigma\frac{h^{\mathrm{dr}}_1}{2\pi}(1-n)(h'_k)^{\mathrm{dr}} = \frac{\sigma(h'_k)^{\mathrm{dr}}}{2\pi}\frac{\partial}{\partial\beta^1}n, \tag{52}$$

where $\alpha = -\frac{1}{2}\sin\omega$, and we used $h_{k+1} = \alpha h'_k$ and $h_1 = \alpha p'$ in the XXZ chain [48]. This then further implies

$$n\frac{\partial}{\partial\beta_{k+1}}\rho^{\mathrm{tot}} = \frac{\sigma n}{2\pi}\frac{\partial}{\partial\beta^1}\rho^{\mathrm{tot}}. \tag{53}$$

Combining these (51) is confirmed, yielding a relation that generalizes the usual case

$$\frac{\partial}{\partial\beta_{k+1}}[n_a(\bar{h}_{k,a} - (h'_{k,a})^{\mathrm{dr}})] = 0, \tag{54}$$

where we restored the string indices. The rest of arguments are the same as before: we take either $\beta_{k+1} \to \pm\infty$, making sure that $n_a(\theta) \to 0$ under the limit for each string species $a$. Assuming the good behavior of $\bar{h}_{k,a}(\theta)$, i.e. $n_a(\theta)\bar{h}_{k,a}(\theta) \to 0$ when $\beta_{k+1} \to \infty$ or $\beta_{k+1} \to -\infty$, the sought statement $\bar{h}_k(\theta) = (h'_k)^{\mathrm{dr}}(\theta)$ is shown.

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
