# Peer review of "Collision rate ansatz for quantum integrable systems"

_SciPost Physics_

## Round 2 · Referee Report · Anonymous (Referee 1) · 2020-6-29

Report

This is an important contribution to the generalized hydrodynamics (GHD) of integrable systems. It provides a rather general mathematical underpinning of the "collision rate Ansatz" which is central to GHD: the Ansatz follows from the existence of a conserved current. Although a very similar argument (applied to the Toda chain) has already appeared in an earlier paper by the second author, this paper discussed it in more intricate settings (e.g. XXZ), and also discussed the boost operators as a generating mechanism of a conserved current. Scientifically, the manuscript is well-written, and deserves prompt publication.

---

## Round 2 · Referee Report · Anonymous (Referee 2) · 2020-7-6

Strengths

1- the paper provides a simple proof of the collision rate ansatz 2- it highlights the connection of the expression of the average currents with the existence of self-conserved currents and the boost operators 3- the discussion is general being applicable to systems with galilean, lorentzian and even on the lattice

Weaknesses

1- it is unclear how to extend the proof to models where the boost operator does not exist

Report

This paper presents a proof of the collision rate ansatz by employing some simple general properties of integrable models: the symmetry of the current-charge susceptibility and the existence of a self-conserved current, which in turns is related to the existence of a boost operator in the theory. Given these ingredients, the authors present a straightforward argument to justify the expression of the average currents on a GGE state.

The paper is interesting and well-written and deserves publication.

Requested changes

1- the fact that the presence of the boost operator implies the existence of a self-conserved current had already been observed (see for instance https://arxiv.org/pdf/1407.1325.pdf). The authors should acknowledge these previous results.

2- the section about the XXZ spin chain seems to repeat the argument reported before. Could the authors clarify if a different proof is needed or if the previous argument could already be applied to the spin chain (and if so why are they repeating it)?

3- what is the status of the collision rate ansatz in models where the boost operator does not exist and no self-conserved current is known, e.g. the Fermi Hubbard model? Do they expect it to hold in any case? Is it therefore more general than the proof presented here ? A comment about this would be appreciated.

---

## Round 2 · Referee Report · Benjamin Doyon (Referee 3) · 2020-7-8

Strengths

1- general method for evaluating current expectation values, valid for integrable / non-integrable, classical / quantum models, without needing technical details

Weaknesses

1- a step seems to be missing an argument, which should be better explained for completing the derivation.

Report

In this paper, the authors find a new method in order to evaluate exact expectation values of currents in integrable models. The method uses a known symmetry property of the so-called B matrix in statistical fluid theory, the matrix of current susceptibilities. With the knowledge of expectation values a single current in all GGEs, the symmetry is sufficient to derive the expectation values of all currents in all GGEs. This is because the symmetry combined with susceptibility equation gives a differential equation for all currents with known "source" term, which can be solved. The method is very general, and applies whenever a current is known. The authors concentrate on the cases where there is Galilean or relativistic invariance, where the particle or energy current is known as it is equal to the momentum density, and other such cases where a current observable is, at the level of microscopic definition, equal to a conserved density. Importantly, this includes the XXZ model.

Although the exact current formula has been known now for some time, and a lot of evidence for its correctness exist, including quite rigorous derivations in the Bethe ansatz framework of B. Pozsgay et al (cited in the paper), given the interest in GHD for this formula it is indeed important to have a new perspective. The method given here is very simple, and extremely general - it bypasses all details of the Bethe ansatz or other techniques for any specific type of integrable model (quantum or classical), being based on a fundamental symmetry relation. Besides the conceptual interest, the question of rigour is quite important. The symmetry relation of the B matrix can be established in a rigorous fashion within any mathematical framework for many-body models, such as the C-star algebra framework - it only needs the susceptibility equation (derivative wrt beta = integrate two-point function), something which is indeed established rigorously in any state of one-dimensional quantum chains at nonzero temperature (wrt any local hamiltonian). Thus, the main non-rigorous part of derivation is the part that relies on the TBA framework.

However, I find that there is one step in the derivation which is unclear. This missing steps reduces the rigour of the derivation, and in fact, I think, makes it incomplete. The claim made by the authors that the only requirement, for the method to work, is for a current observable to be equal to a conserved density, is I believe not entirely true. Given that the main point of the paper is to present a new method for a known formula, it is important for the authors to fully clarify this step, or to clarify the fact that this is a missing step.

Thus, I will be very happy to accept the paper for publication, once the missing step described below is either clarified - i.e. details are provided for its correctness, or the authors explain my misunderstanding - or it is emphasised in the paper that this is a missing step where physical intuition is required.

I would also ask the authors to add some references, relevant to the problem they are discussing, and to clarify the small additional things below.

The missing step is as follows

I don't think the equality $\bar v = v^{\rm eff}$ has been established: the presence of a self-conserved current is not entirely sufficient. Another requirement for the method to work is a known value of all currents for a specific value of the Lagrange parameter associated to the current. This will form the initial condition for the differential equation obtained by the symmetry of the B matrix. This is not so obvious, and I think this is a point where the authors' argument fails - although in many cases, I believe physical arguments can be supplemented.

That is:

  • In the Galilean case, page 5, it is indeed established that the derivative wrt $mu_0$ is zero. However, more is needed in order to establish that the constant [which of course may depend on all $mu_i$ for $i$ nonzero] is zero. The case $mu_0 \to\infty$ is correctly taken, but the possibility remains that $\bar v$ grows with $mu_0$, and thus that the constant is nonzero. One needs to provide stronger arguments, probably the vanishing of all currents in this limit. In this Galilean case, it is clear, at least on physical grounds, that any current must vanish as $\mu_0\to\infty$, as there remains no particle. This can be used as an argument, however I do not know how rigorous it is; it is surely possible to bound the current as the potential is made large in a more or less rigorous fashion. The authors should discuss this.

  • In the relativistic case, page 6, the same problem arises; in principle, $\bar v$ may grow as $\mu_2\to\infty$. Here, however, it is more tricky. One takes the limit $\mu_2\to\infty$. If all other $\mu_i$ are simultaneously taken to infinity (is this allowed?), this is the zero temperature limit. Can we then say, at least on physical grounds, that all currents vanish? What about if the other $\mu_i$'s are kept fixed - this is not a zero-temperature limit, what is the argument for the vanishing of the currents?

  • the same problem arises in the XXZ model, page 8.

Small additional things

“Existence of boost operator” this is not so clear to me: in any theory, we can consider the first moment of a conserved density, this always exist, but here there is something more that is said. Please could you clarify the discussion?

Conclusion: I wouldn't say that the existence of a self-conserved current being related to a boost operator is surprising - this is rather known phenomenon in integrable systems, and of course it is the definition of Lorentz / Galilean invariance that the current of energy / particle is the (conserved) momentum density.

Page 7: integral operators are already used around eq 6

Additional refs

1) effective velocity, currents

As this paper is about the effective velocity, I think it is important to relate as much as possible the full literature on this quantity.

The expression first appeared in:

L. Bonnes, F. H. L. Essler, A. M. L\"auchli, 'Light-cone' dynamics after quantum quenches in spin chains, {\em Phys. Rev. Lett.} 113, 187203 (2014)

There, it was in the context of an expression for a current; nevertheless, the physics was very much related, so I think this is a relevant reference.

In addition to [29-31], three other references should be mentioned, where this formula is established in various settings and with various level of rigour. I think this is particularly important, in order to put the work in its context. The works are:

M. Fagotti, Charges and currents in quantum spin chains: late-time dynamics and spontaneous currents, J. Phys. A 50, 034005 (2017).

There I believe the formula is established in models with free-fermion structure. I think this is quite rigorous; of course the free-fermion case is much more "trivial", but still important.

A. Urichuk, Y. Oez, A. Klumper and J. Sirker, The spin Drude weight of the XXZ chain and generalized hydrody-namics, SciPost Phys. 6, 5 (2019).

There it is derived for the spin current, the method is rather general for models with spin conservation, and also quite rigorous. However, it is limited to spin currents.

J. De Nardis, D. Bernard and B. Doyon, Diffusion in generalized hydrodynamics and quasi- particle scattering, SciPost Phys. 6, 049 (2019).

There the formula for generic currents is obtained from finite-density form factors, in the Lieb-Liniger model. Again the method is quite general, relatively simple, but much less rigorous, based on a form factor theory that is not yet well developed.

In addition, the author might want to cite (as they wish):

Z. Bajnok and I. Vona, Exact finite volume expectation values of conserved currents, Phys. Lett. B 805, 135446 (2020)

where finite-volume expressions are considered.

Finally, although this appeared after the present work, if they want (no obligation) they may cite the recent work

B. Pozsgay, Algebraic construction of current operators in integrable spin chains, arXiv:2005.06242 (2020).

2) B matrix

Ref 23 indeed presents a quite general proof, and should indeed be mentioned, but a proof was already presented in [5], which also should be mentioned. I think it is worth also citing

D. Karevski, G.M. Schütz, Charge-current correlation equalities for quantum systems far from equilibrium, SciPost Phys. 6, 068 (2019)

where a quite extensive study is made of this relation, showing the general context where it holds.

3) Drude weight in XXZ

There are many more references to this, this has quite a long history.... I don't want to "impose" any specific list of references, but maybe the authors can have a look, for instance in the introduction of ref [21] B. Doyon and H. Spohn, Drude weight for the Lieb-Liniger Bose Gas, SciPost Phys. 3, 039 (2017) there is a quick historical discussion (not necessarily the best!); or the recent review arXiv:2003.03334, Finite-temperature transport in one-dimensional quantum lattice models, B. Bertini, F. Heidrich-Meisner, C. Karrasch, T. Prosen, R. Steinigeweg, M. Znidaric

Requested changes

1- clarify the step mentioned in main report 2- clarify the small things mentioned in main report 3- add references mentioned in main report

---

## Editorial Decision

resubmitted